# ICA MODEL ESTIMATION USING AN OPTIMIZED VERSION OF GENETIC ALGORITHMS

## ABSTRACT

This paper presents a method of estimating the independent component analysis model based on the use of a training algorithm based on an optimized version of genetic algorithms with a neural network algorithm. The mixed training algorithm is applied to optimize the objective function negentropy used to estimate the ICA model. The proposed estimation algorithm improves the training scheme based on genetic algorithms by using for crossover the most suitable chromosomes evaluated by the objective function with the parameters calculated calculated accordingly by a multilayer neural network algorithm. The performances of the proposed algorithm for estimating the independent components were evaluated through a comparative analysis with the versions of FastICA algorithms based on the standard Newton method, as well as on the secant method of derivation of the training scheme at the level of the optimization stage of the approximate objective function. The experimental results for the proposed algorithm for estimating the independent components are established in specific blind source separation applications using unidimensional and bidimensional signals.

## 1 INTRODUCTION

In various research fields such as signal processing, mathematical statistics or neural networks, an important problem is the need to obtain adequate representations for multidimensional data. The problem can be formulated in terms of finding a function $f$ such that the $n$ dimensional transform defined by $s = f(x)$ possesses some desired properties, where $x$ is a $m$ dimensional random vector. Being given its computational simplicity, frequently the linear approach is attractive, that is the transform is

$$s = Wx \tag{1}$$

where $W$ is a matrix to be optimally determined from the point of view of a pre-established criterion. There are a long series of methods and principles already proposed in the literature for solving the problem of fitting an adequate linear transform for multidimensional data as for instance, Principal Component Analysis (PCA), factor analysis, projection methods and Independent Component Analysis (3), (6), (18), (19). The aim of Independent Component Analysis is to determine a transform such that the components $s_i, i = \overline{1..n}$ becomes statistically independent, or at least almost statistically independent. In order to find a suitable linear transform to assure that equation 1 $s_i, i = \overline{1..n}$ become 'nearly' statistically independent several methods have been developed so far. Some of them, as for instance Principal Component Analysis and factor analysis are second order approaches, that is they use exclusively the information contained by the covariance matrix of the random vector $x$, some of them, as for instance the projection methods and blind deconvolution are higher order methods that use an additional information to reduce the redundancies in the data. Independent Component Analysis has became one of the most promising approaches in this respect and, consists in the estimation of the generative model of the form $x = As$, where the $s = (s_1 s_2, \ldots s_n)^T$ are supposed to be independent, and $A$ is the mixing matrix $m \times n-$ dimensional of the model. The data model estimation in the framework of independent component analysis is stated in terms of a variational problem formulated on a given objective function (5), (24). The aim of the research reported in this paper is to introduce a new version of the FastICA algorithm; an algorithm that is based on genetic algorithms and artificial neural networks and to analyze the performances of the algorithm in blind signal separation applications.

## 2 DERIVATION OF TRAINING ALGORITHM FOR INDEPENDENT COMPONENT ESTIMATION BASED ON GENETIC ALGORITHMS AND NEURAL NETWORKS

### 2.1 THE STANDARD FASTICA ALGORITHM

In this part the ICA model and the standard FastICA algorithm are briefly exposed. The ICA model is state as $x = As$, where $x$ is the observations vector and $A$ is the mixing matrix of the original sources, $s$. The aim is to determine the sources, on the basis of $x$. One of the basic working assumption in estimation the ICA model is that the sources $s$ are statistically independent and they have nongaussian distributions. This way the problem becomes to find the weighting matrix $W$ (the demixing matrix), such that the transform $y = Wx$ gives suitable approximations of the independent sources. In the following, the standard version of the independent components estimation algorithm using negentropy as objective function is presented (18). The negentropy is defined by:

$$I(y) = H(y_{gauss}) - H(y) \tag{2}$$

where $H(y) = - \int p_y(\eta) \log p_y(\eta) d\eta$ is the differential entropy of the random vector $y$. Considering that that the Gaussian repartition is of largest differential entropy in the class of the repartitions having the same covariance matrix, the maximization of the negentropy (2) gives the best estimation of the ICA model. Although this approaches is well founded from information point of view the direct use of the expression (2) is not computationally tractable, and some approximations are needed instead. We use the approximation introduced in (Hyvarinen, 98):

$$I(y) = [E\left\{G(y)\right\} - E\left\{G(\nu)\right\}]^2 \tag{3}$$

where $G$ is an nonquadratic function, $\nu$ and $y$ are Gaussian variables of zero mean and unit variance. Some of the most frequently used expressions of $G$ are, $G_1(y) = \frac{1}{a_1} \log cosh(a_1 y); \ 1 \leq a_1 \leq 2$, $G_2(y) = -\exp(-\frac{y^2}{2}); \ G_3(y) = \frac{y^4}{4}$. Note that the expressions of their first order derivatives are given by: $g_1(y) = \tanh(a_1 y); \ g_2(y) = y\exp(-\frac{y^2}{2}); \ g_3(y) = y^3$, respectively. The variational problem can be formulated as a constraint optimization problem as follows,

$$\max F(w), \ \|w\|^2 = 1 \tag{4}$$

that is the objective function $F(w)$ has to be maximized on the unit sphere. In case the negentropy is taken as the objective function, we get,

$$F(w) = [E\left\{G(y)\right\} - E\left\{G(\nu)\right\}]^2 \tag{5}$$

where $y = w^T z$. To solve the optimization problem from the equation 4 relation we write the Lagrange function using the Lagrange multiplications method:

$$L(w) = F(w) - \lambda(\|w\|^2 - 1) \tag{6}$$

The weighting vectors being normalized, we arrive at the following approximation scheme,

1. $w \leftarrow E\{zg\left(w^T z\right)\} - E\{g^{'}\left(w^T z\right)\}w$
2. $w \leftarrow w/\|w\|$.

A detailed version of the FastICA algorithm based on Lagrange function and Newton method is described in the following scheme.

**The standard FastICA algorithm based on Lagrange function and Newton method**

Step 1 : Center the data to make its mean zero.

Step 2 : Applying the whitening transformation and obtaining the data signals $z$.

Step 3 : Choose the number of independent components $n$ and set counter $r \leftarrow 1$.

Step 4 : Choose an initial approximation (usually randomly generated subunit values) of unit norm for $w_r$.

Step 5 : Let

$$w \leftarrow E\{zg\left(w^T z\right)\} - E\{g^{'}\left(w^T z\right)\}w \tag{7}$$

where $g$ is previously defined ($versions$ : $g_1(y) = \tanh(a_1 y)$; $g_2(y) = y \exp(-\frac{y^2}{2})$; $g_3(y) = y^3$).

Step 6 : Do the orthogonalization transform:

$$w_r \leftarrow w_r - \sum_{j=1}^{r-1}(w_r^T w_j)w_j \tag{8}$$

Step 7 : Let $w_r \leftarrow w_r/\|w_r\|$.

Step 8 : If $w_r$ has not converged ($< w_{n+1}, w_n > \nrightarrow 1$), go back to step 5.

Step 9 : Set $r \leftarrow r + 1$. If $r \leq n$ then go to step 4.

Step 10 : The discovered signals are determined based on the weight vectors $w_1, ..., w_n$ as follows: $s\_discovered_i = w_i{}^T * z$, for $i = \overline{1..n}$.

## 2.2 GENETIC ALGORITHMS

Genetic algorithms are adaptive heuristic search techniques based on principles genetics and natural selection, stated by Darwin (survival of the best adapted). The mechanism is similar to the biological process of evolution. According to this process only the species that adapt better to the environment are able to survive and evolve over generations, while those less adapted do not manage to survive and eventually disappear, as a result of natural selection. The probability that the species will survive and evolve over generations becomes greater as the degree of adaptation increases, which in terms of optimization means that the solution approaches the optimum (17), (21). As practical applications, genetic algorithms are most often used in solving problems of optimization, planning or search. Evolutionary algorithms use a vocabulary borrowed from genetics: the evolution is simulated through a sequence of generations of a population of candidate solutions; a candidate solution is called a chromosome and is represented as a string of genes; the population evolves through the application of genetic operators: selection, mutation and crossing; the chromosome on which a genetic operator is applied is called the parent and the chromosome the result is called the descendant; selection is the process by which the chromosomes that will survive in the generation are chosen next; better adapted individuals will be given greater chances; the degree of adaptation to the environment is measured by the fitness function; the solution returned by a genetic algorithm is the best individual from the last generation. The *fitness function* is used to measure the quality of the chromosomes. She is formulated starting from the numerical function to be optimized. *The general structure of a genetic algorithm* Set the time $t \leftarrow 0$. Creation of the initial population $P(t)$. Evaluation of the initial population with the fitness function ($fitness(P(t))$). While the final condition is false: $t \leftarrow t + 1$, Selection of new generation $P(t)$ from $P(t - 1)$, Application of the crossover operator for the selected chromosomes for the new population $P(t)$, Evaluation of the new population with the fitness function ($fitness(P(t))$) and determining the final chromosomes (keeping the best chromosomes).

## 2.3 THE FASTICA ALGORITHM VERSION BASED ON THE GENETIC ALGORITHMS AND ARTIFICIAL NEURAL NETWORKS

This part for the ICA model describes the FastICA type algorithm developed by using genetic algorithms to optimize the negentropy objective function, as well as artificial neural networks in the process of crossing the genes of the selected chromosomes to produce the new generation of chromosomes. In the following, the numerical estimation of the independent components is going to be obtained using the genetic algorithms and artificial neural networks approaches, the variational problem being imposed on the negentropy taken as criterion function. The variational problem can be formulated as a constraint optimization problem as follows,

$$\max F(w), \ \|w\|^2 = 1 \tag{9}$$

that is the objective function $F(w)$ has to be maximized on the unit sphere. In case the negentropy is taken as the objective function, we get,

$$F(w) = [E\{G(y)\} - E\{G(\nu)\}]^2 \tag{10}$$

where $y = w^T z$. To solve the optimization problem from the equation 9 relation a genetic algorithm will be used to maximize the objective function $F(w)$. In the stages of transformation of the chromosomes in the current population, by using genetic operators, a artificial neural networks model will be used for crossing the genes of the chromosomes selected for reproduction. For the proposed model, the optimization of the objective function $F(w)$) is achieved by using a genetic algorithm in which the roulette type method is used in the selection stage of candidate chromosomes for the new generation, in the reproduction stage the intermediate recombination method combined with a model is used the backpropagation algorithm from artificial neural networks, with only one hidden layer in the neural architecture and 12 neurons on the hidden layer, that calculates the appropriate values of the recombination weights by referring to the performance standard achieved in the previous generation by the best chromosome evaluated by the fitness function, and in the mutation step the Gaussian mutation model is used. Looking at the crossing stage, the combined method with artificial neural networks (*CMCNN learning algorithm* - combined method for crossover with artificial neural networks) can be formulated as follows:

1) for each $W_i$ chromosome from the $N$ chromosomes selected in the selection stage, in order to create the new generation, $N - 1$ artificial neural networks (denoted by $NW_{i1}$, $NW_{i2}$, ..., $NW_{iN-1}$) corresponding to the other $N - 1$ chromosomes initially chosen are designed and developed.

2) the artificial neural networks $NW_{i1}$, $NW_{i2}$, ..., $NW_{iN-1}$ establish a learning process of the memory weights associated with the intermediate recombination weights from the reproduction stage of the genetic algorithm used.

3) the training law for the $NW_{ij}$ neural network is based on the backpropagation training algorithm with the target value returned by the objective function value for the best chromosome from the previous generation of chromosomes.

4) Finally, the values of the synaptic weights corresponding to the neural network $NW_{ij}$ will be chosen for which the best approximation of the target value is obtained. The neural network $NW_{ij}$ provides the following information: the values of the synaptic weights represent the most suitable values for the weights used in the crossover stage of the genes of the $W_i$ chromosome, and the $W_j$ chromosome is the most suitable for the crossover with $W_i$ in order to create new chromosomes for the next generation.

A detailed version of the FastICA algorithm based on Genetic Algorithms and Artificial Neural Networks is described in the following scheme.

**The standard FastICA algorithm based on Genetic Algorithms and Artificial Neural Networks (ICAGNN)**

1 : Center the data to make its mean zero.

2 : Applying the whitening transformation and obtaining the data signals $z$.

3 : Choose the number of independent components $n$ and set counter $r \leftarrow 1$.

4 : Choose an initial approximation (usually randomly generated subunit values) of unit norm for $w_r$.

5 : The weight vector $w_r$ is determined by applying the genetic algorithm (GA).

    5.1 : Set the time $t \leftarrow 0$.

    5.2 : creation of the initial population $P(t) = \{W_1, ..., W_M\}$ for the weight vector $w_r$ and $M$ is the number of chromosomes.

    5.3 : Evaluation of the initial population with the fitness function $F(w)$.

    5.4 : While the final condition is false (the convergence condition is rendered by a maximum number of generations or insignificant updates between epochs):

        $a.\ t \leftarrow t + 1$.

$b$. Apply the chromosome selection operator that participates in the formation of the new generation of chromosomes by the roulette method. The set of chromosomes selected, by renumbering, are $SP(t) = \{W_1, ..., W_M\}$, with $M < N$, $SP$ represents the selected chromosomes.

$c$. Application of the *CMCNN learning algorithm* for the selected chromosomes $SP(t)$. The target value is set by the objective function value for the best chromosome in the previous generation. The obtained values of the synaptic weights represent the most suitable values for the weights used in the crossover stage of the genes of the $W_i$ chromosome, and the $W_j$ chromosome is the most suitable for the crossover with $W_i$ in order to create new chromosomes for the next generation.

$d$. New descendant chromosomes are determined by crossing the genes of the corresponding $W_i$ and $W_j$ chromosomes by using the crossing weights resulting from the *CMCNN learning algorithm.*

$e$. The mutation operator in the Gaussian variant is applied for some chromosomes resulting from crossing.

$f$. Do the orthogonalization transform for the obtained chromosomes:

$$W_s \leftarrow W_s - \sum_{j=1}^{r-1}(W_s^T w_j)w_j \tag{11}$$

with $s = 1..M$.

$g$. Evaluation of the chromosomes population (new descendant chromosomes resulting after crossing over, mutation and orthogonalized and chromosomes from the current population $P(t-1)$) with the fitness function $F(w)$ and determining the final chromosomes (eg keeping the best chromosomes).

5.5 : The best chromosome from the last generation from the point of view of the fitness function will denote the weight vector $w_r$ set for the $r$ independent component.

6 : Let $w_r \leftarrow w_r/\|w_r\|$.

7 : Set $r \leftarrow r + 1$. If $r \leq n$ then go to step 4.

8 : The discovered signals are determined based on the weight vectors $w_1, ..., w_n$ as follows: $s\_discovered_i = w_i^T * z$, for $i = \overline{1..n}$.

9 : Evaluation of the error that occurs in the process of discovering the original signals from the mixed signals for the $n$ independent components (the absolute distance type error function calculated for the discovered independent signals and the original ones will be used).

## 3 RELATED WORK

The development of the field of independent components analysis led to the appearance of several works that bring improvements to the standard FastICA algorithm for determining the original sources from mixtures of multidimensional input data. J.M. Gorriz and C.G. Puntonet proposed a genetic algorithm to minimize a non-convex and nonlinear cost function based on statistical estimators for solving the independent component analysis problem of blind source separation (12). H. Yaping, L. Siwei proposed a new ICA feature selection based on genetic algorithms. To demonstrate its effectiveness, recognition experiments were conducted for face and iris recognition (36). Also, H. Shahamat and A.A. Pouyan proposed a new method for classifying subjects into schizophrenia and control groups using functional magnetic resonance imaging (fMRI) data. In the preprocessing step, the number of fMRI time points was reduced using principal component analysis. Then, independent component analysis was used for further data analysis and for feature selection, genetic algorithm was used to obtain a set of features with high discriminating power (31).

## 4 EXPERIMENTAL ANALYSIS

To evaluate the performance of the proposed algorithm based on genetic algorithms and neural networks, three categories of input data (input signals) are considered. Each category of data includes signals generated with a certain probability distribution or two-dimensional signals. Thus,

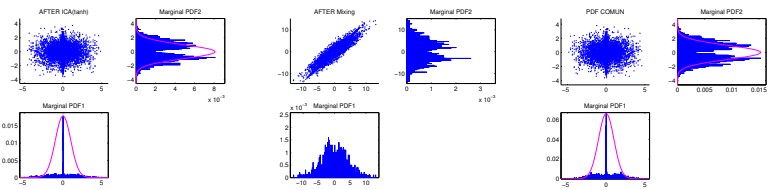

Figure 1: Test I - 2 normal signals (density function of Discovered signals, Mixed Signals and Original Signals)

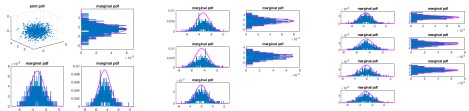

Figure 2: Test I - data density functions used in cases with 3, 5 and 7 signals with normal distribution

the three categories include data generated with uniform distribution, normal distribution, respectively, face images. For each category of data, a comparative study is carried out regarding the performance of determining the independent components with a variation in the number of input data (the cases with 2 mixed signals, 3 mixed signals, 5 mixed signals, respectively, 7 mixed signals for each data category are analyzed signals). Also, for each category of data and each number of mixed input signals, the experimental results are established both for the proposed based algorithm (ICAGNN) and for the versions of the standard FastICA algorithm based on the use of the negentropy objective function optimization through the Lagrange method and Newton or secant numerical methods. For the population of chromosomes used by the genetic algorithm, between 20-50 chromosomes were gradually used, and the size of each chromosome is represented by the size of the representation space of the mixed input signals for the algorithm. In order to establish the performances in the discovery of independent signals from mixed signals, the average absolute error at the level of the values of the compared signals was used, and also the average value of the number of iterations required for the convergence of the weight vectors associated with the discovered independent signals was also retained. The mean absolute error (AbsME) used is expressed by: $AbsME = \sum_{i=1}^{N} |s_i - s\_estimated_i|/N$, where $s_i$ and $s\_estimated_i$ represent the $i$-th pixel values for original and restored signals, respectively, and $N$ is the total number of pixels. For the negentropy objective function, use the proposed algorithm for estimating the independent components, the approximations established in the description of the negentropy are used, being established experimentally for each choice of the function $g$ from the approximation formula. From the comparative study carried out for the three categories of one-dimensional and two-dimensional signals mixed with a variable number of input data, the proposed algorithm (ICAGNN) based on genetic algorithms and artificial neural networks is distinguished by an improved rate of the average of the iterations required for the convergence of the weight vectors associated with the recognized independent signals, having a quality of the discovered signals similar to the performances of the standard FastICA estimation algorithms based on the secant method or the Newton method.

Table 1: The convergence rate of algorithms for estimating independent components from 2 mixed signals for all experimental tests

| Algorithm | The convergence rate for 2 signals for each case I, II, III | | | | | | | | |
|---|---|---|---|---|---|---|---|---|---|
| No. of Test | *tanh* | *exp* | *kurt* | *tanh* | *exp* | *kurt* | *tanh* | *exp* | *kurt* |
| Newton | 3.0 | 2.5 | 3.5 | 2.5 | 3.0 | 3.0 | 4.0 | 3.5 | 3.5 |
| Secant | 2.5 | 2.5 | 4.0 | 3.0 | 2.5 | 3.0 | 4.5 | 3.0 | 4.5 |
| ICAGNN | 2.5 | 2.0 | 3.0 | 2.5 | 2.5 | 3.0 | 2.5 | 3.0 | 3.0 |

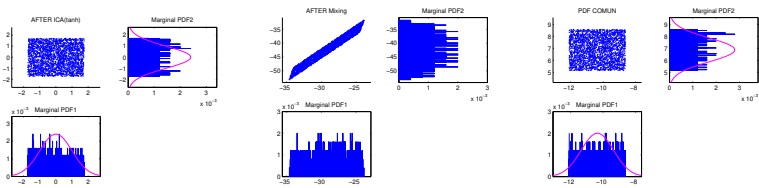

Figure 3: Test II - 2 uniform signals (density function of Discovered signals, Mixed Signals and Original Signals)

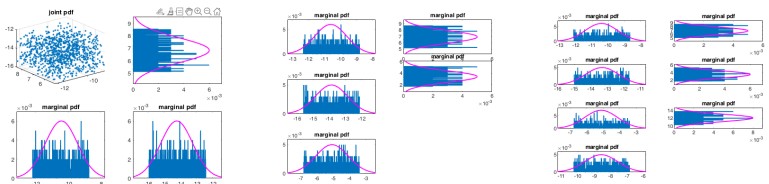

Figure 4: Test II - data density functions used in cases with 3, 5 and 7 signals with uniform distribution

## 4.1 DESCRIPTION OF THE EXPERIMENTAL TESTS IN THE CASE OF UNIDIMENSIONAL SIGNALS

For the unidimensional case, two experimental tests are established by using signals generated with a normal, respectively uniform, distribution.

Test I. For this experimental test, the input signals are generated according to the normal distribution with implementation in the Matlab language. For this category of input data, a number of 2, 3, 5, respectively, 7 mixed signals is used, and the performances in discovering the original independent signals is determined, both for the proposed algorithm and for the algorithms used in the comparative study. The signals used as input data for establishing mixed signals and determining independent signals by the proposed algorithm are represented in figures 1 - 2 for each case of the number of mixed input data.

Test II. For this experimental test, the input signals are generated according to the uniform distribution with implementation in the Matlab language. For this category of input data, a number of 2, 3, 5, respectively, 7 mixed signals is used, and the performances in discovering the original independent signals is determined, both for the proposed algorithm and for the algorithms used in the comparative study. The signals used as input data for establishing mixed signals and determining independent signals by the proposed algorithm are represented in figures 3 - 4 for each case of the number of mixed input data.

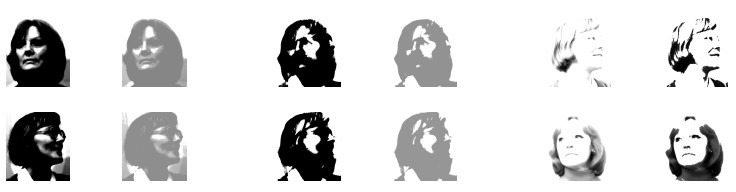

Figure 5: Test III - 2 images of faces (Discovered faces, Mixed faces and Original faces)

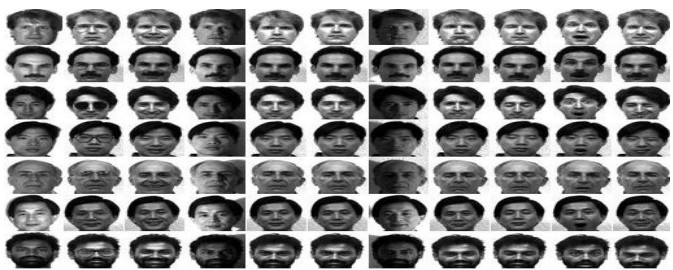

Figure 6: Test III - images of faces used in cases with 3 (first 3 faces), 5 (first 5 faces) and 7 faces from Yale database with with 11 associated images each

Table 2: The convergence rate of algorithms for estimating independent components from 3 mixed signals for all experimental tests

| Algorithm | The convergence rate for $3$ signals for each case I, II, III | | | | | | | | |
|---|---|---|---|---|---|---|---|---|---|
| No. of Test | *tanh* | *exp* | *kurt* | *tanh* | *exp* | *kurt* | *tanh* | *exp* | *kurt* |
| Newton | 4.0 | 3.66 | 4.33 | 3.66 | 3.0 | 3.66 | 4.0 | 4.33 | 5.33 |
| Secant | 4.0 | 3.33 | 4.66 | 4.00 | 3.66 | 3.66 | 4.33 | 4.00 | 4.66 |
| ICAGNN | 3.00 | 3.0 | 3.66 | 2.66 | 3.00 | 3.33 | 2.66 | 3.0 | 2.66 |

## 4.2 DESCRIPTION OF THE EXPERIMENTAL TESTS IN THE CASE OF BIDIMENSIONAL SIGNALS

Test III. For this experimental test, the input signals are represented by the images of faces. For this category of input data, a number of 2, 3, 5, respectively, 7 mixed faces is used, and the performances in discovering the original independent faces is determined, both for the proposed algorithm and for the standard algorithms used in the comparative study. Thus, by using a different number of two-dimensional image signals, the mixed images are established, and the proposed algorithm is applied versus the FastICA algorithm for the identification of independent image sources. The faces signals used as input data for establishing mixed faces and determining independent faces by the proposed algorithm are represented in figures 5 - 6 for each case of the number of mixed input data.

## 4.3 THE RESULTS OF THE COMPARATIVE STUDY ON THE CONVERGENCE RATE

In this part, the performance results in the detection of hidden signals are presented for the proposed algorithm based on genetic algorithms and neural networks (ICAGNN) in comparison with standard algorithms for estimating independent components. If in terms of the quality of the discovered signals evaluated by the absolute error function it shows similar values for all the algorithms used, in the analysis of the convergence rates, the proposed algorithm ICAGNN stands out for improved values relative to the number of iterations required to establish the weight vectors associated with the independent signals recognized by each algorithm from the comparative study. The results obtained for the estimation of the independent components by applying the proposed ICAGNN algorithm compared to the FastICA algorithms, are shown in tables I-IV corresponding to 2, 3, 5, 7 respectively mixed signals. Also, the mean values for the number of iterations required for the convergence of the weight vectors are presented for the three choices of the function $g$ (versions $g_1$, $g_2$ and $g_3$) in the approximation of the negentropy objective function used in the estimation of the independent components.

## 5 SUMMARY AND CONCLUSIONS

In the work carried out, a new version of the FastICA-type independent components estimation algorithm was developed by optimizing the negentropy objective function by using genetic algorithms in an updated form by using appropriate weights in the crossover stage resulting from the application of a neural network of multilayer backpropagation type. The performance of the proposed algorithm was determined in a study compared to the standard version of the estimation algorithm based on the

Table 3: The convergence rate of algorithms for estimating independent components from 5 mixed signals for all experimental tests

| Algorithm | The convergence rate for 5 signals for each case I, II, III | | | | | | | | |
|---|---|---|---|---|---|---|---|---|---|
| No. of Test | *tanh* | *exp* | *kurt* | *tanh* | *exp* | *kurt* | *tanh* | *exp* | *kurt* |
| Newton | 8.40 | 7.20 | 8.40 | 4.60 | 5.00 | 5.40 | 7.80 | 8.40 | 7.80 |
| Secant | 7.80 | 7.23 | 8.40 | 4.80 | 5.40 | 5.80 | 7.40 | 7.20 | 8.00 |
| ICAGNN | 6.20 | 6.00 | 7.20 | 3.40 | 3.80 | 4.80 | 4.80 | 5.00 | 4.80 |

Table 4: The convergence rate of algorithms for estimating independent components from 7 mixed signals for all experimental tests

| Algorithm | The convergence rate for 7 signals for each case I, II, III | | | | | | | | |
|---|---|---|---|---|---|---|---|---|---|
| No. of Test | *tanh* | *exp* | *kurt* | *tanh* | *exp* | *kurt* | *tanh* | *exp* | *kurt* |
| Newton | 8.42 | 8.14 | 8.85 | 5.28 | 6.28 | 5.57 | 8.57 | 8.42 | 8.28 |
| Secant | 8.14 | 8.42 | 8.85 | 5.28 | 5.57 | 6.85 ) | 8.14 | 8.42 | 8.85 |
| ICAGNN | 5.28 | 5.57 | 6.85 | 3.85 | 4.28 | 5.57 | 4.28 | 5.14 | 5.28 |

Newton method, as well as to the version developed based on the secant method. The experimental stability results for several types of signals and for several cases regarding the volume of input data, showed good performance in the estimation of independent signals both from the point of view of the quality of the original sources, and especially from the point of view of the convergence rate improved for the proposed algorithm.

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
