# OpenReview forum: "ICA model estimation using an optimized version of genetic algorithms"
_ICLR.cc/2024/Conference — Submitted to ICLR 2024_

### Official Review · Reviewer_t4Z3 · 2023-10-30

**Soundness:** 2 fair
**Presentation:** 2 fair
**Contribution:** 2 fair
**Rating:** 6
**Confidence:** 2

**Summary:**

This paper proposed a mixed algorithm based on a genetic algorithm and neural network for ICA model estimation. After the algorithm descriptions, some experiments are provided for verification.

**Strengths:**

Using other algorithms for the ICA model estimation may be a new research direction

**Weaknesses:**

I am not familiar with ICA, so I cannot exactly evaluate how much the contribution of the proposed work in this paper to the community of ICA. However, in my personal view, there is much new thing in using genetic algorithms with neural networks for optimization tasks. Compared to the traditional methods, the used methods are often not effective, and the results are also not always with the same values. Unfortunately, I did not see such kinds of discussion.

**Questions:**

See above

---

### Official Review · Reviewer_ZRC8 · 2023-10-31

**Soundness:** 1 poor
**Presentation:** 1 poor
**Contribution:** 1 poor
**Rating:** 3
**Confidence:** 3

**Summary:**

This paper proposes a genetic algorithm and neural network-based optimization method for independent component analysis. The loss function used in the proposed method is the same one in the FastICA algorithm. The performance of the proposed method is experimentally compared to the FastICA variants on artificial and face image datasets.

**Strengths:**

- A new genetic algorithm-based optimization method for ICA is presented.

**Weaknesses:**

- While FastICA is a well-known ICA method, there exist a lot of improved algorithms for ICA. The proposed method should be compared to state-of-the-art ICA algorithms and FastICA variants to demonstrate its effectiveness.
- The existing works already attempt to use a genetic algorithm for ICA. For example,
    * G. Wen, C. Zhang, Z. Lin, Z. Shang, H. Wang and Q. Zhang, "Independent component analysis based on genetic algorithms," 2014 10th International Conference on Natural Computation (ICNC), Xiamen, China, 2014, pp. 214-218, doi: 10.1109/ICNC.2014.6975837.
    * H. Azad and M. Hatam, "Maximum likelihood independent component analysis using GA and PSO," 2016 24th Iranian Conference on Electrical Engineering (ICEE), Shiraz, Iran, 2016, pp. 776-781, doi: 10.1109/IranianCEE.2016.7585625.
- The justification and design principle of the proposed method is unclear. In addition, the reason for the performance improvement by the proposed method is also unclear.
- There are many typos and unclear descriptions in the paper.

**Questions:**

- How is the computational cost of the proposed method for ICA optimization compared to other ICA algorithms?
- I cannot identify the role of the neural network in the proposed method. It would be better to clarify the detailed algorithm, including the input/output and training data of the neural network, and the advantages of using the neural network.

---

> ### Author Response · Authors · 2023-12-02
> **Clarifications regarding the work (the justification and design principle of the proposed method, the role of the neural network in the proposed method, the computational cost of the proposed method for ICA, Input/output and training data)**
>
> 1. The justification and design principle of the proposed method:
> -	The design principle is to estimate the ICA model by optimizing the objective function through improved genetic algorithms in terms of the crossover stage of selected chromosomes to form offspring for the new generation. Improvement for GA is achieved in the crossover stage by determining the most suitable weights associated with the linear combination used for linear gene recombination method of selected chromosomes to produce new offspring, through a multilayer neural network (CMCNN algorithm).
> -	The justification is given by the computational advantages (no longer needing to apply the Lagrange multiplier method with the necessary derivatives and solving the multidimensional equation through Newton's or secant method) for determining the training rule that establishes weight vectors associated with discovering independent components.
>
> 2. The role of the neural network in the proposed method:
> -	The role of the neural network is to determine the optimal values for linear recombination of selected chromosomes in order to create a new generation. Using the neural network, we select the recombination weights and the most suitable chromosomes for crossover (the offspring have fitness function values from the genetic algorithm used greater than those of the best chromosomes from the previous generation).
>
> 3. Computational cost of the proposed method for ICA:
> -	Traditional calculation costs of genetic algorithm are mitigated by using neural networks that determine improved values for linear recombination in the crossover stage, thus obtaining offspring with improved fitness function values estimating weight vectors associated with independent components.
> -	In general, the evolutionary algorithm improved by appropriate crossover regarding the offspring of the new generation and determining vectors associated with independent components has a faster convergence rate than standard methods based on Newton or secant numerical methods.
>
> 4. Input/output and training data:
> 	The input data of the algorithm are the recorded mixed signals which are centered and whitened in the preprocessing stages. To determine each independent component, a weight vector Wi is used, which is determined by optimizing the objective function F(Wi) using an evolutionary method with genetic algorithms. The genetic algorithm used has as input prototypes of the weight vector Wi that will discover independent component i. In the evolution process, after selecting chromosomes to participate in creating offspring, in the crossover stage, a multilayer neural network algorithm is applied to weights corresponding to linear combinations used for crossing genes with real values of chromosomes. In the evolution process, the genetic algorithm after the selection of the chromosomes that will participate in the creation of offspring, in the crossover stage the multilayer neural network algorithm is applied regarding the weights corresponding to the linear combination used to cross the genes with real values of the selected chromosomes. The role of the neural network is to provide the most appropriate linear recombination weights of the genes of the selected chromosomes so that the descendants used in the genetic algorithm present improved performance of the fitness function compared to the representatives of the previous generation from the optimization genetic algorithm. The weight vectors finally obtained after optimizing the objective function with genetic algorithms are used to establish the independent components.

---

### Official Review · Reviewer_6jJ9 · 2023-12-12

**Soundness:** 1 poor
**Presentation:** 1 poor
**Contribution:** 1 poor
**Rating:** 1
**Confidence:** 5

**Summary:**

The paper uses a genetic algorithm to achieve ICA. Formally, the search space is that of vectors $w$ on the sphere ($\|w\| = 1$) of dimension $d$, where the dimension of the data is $d$. The objective function is that of the fast ICA method.

**Strengths:**

We are prepared to believe that GAs and stochastic optimization at large might advance the state of the art in problems that are not amenable to mainstream optimization. This paper asks the excellent question of whether GAs can be used to advance the state of the art in Fast ICA,

**Weaknesses:**

The paper does not follow the standard practices for scientific papers, in particular regarding the structure of the paper, the state of the art, the description of the method and the empirical validation.

* The state of the art is antique; I had to number the references manually.

* Description of the method. The authors must motivate their choice of GAs: the FastICA problem being a continuous one, the current prominent approach for stochastic optimization is the CMA-ES approach (https://en.wikipedia.org/wiki/CMA-ES).

* Experimental setting: i) It is all mixed with the presentation of the method. ii) It is not reproducible. Authors indicate "the convergence rate", "the average of the iterations required for the convergence of the weight vectors" "the convergence condition is rendered by a maximum number of generations or insignificant updates between epochs": all this is not reproducible.

Suggestions: The good practice is to give the dimension of the search space, to investigate how the method scales up with the size of the problem, to indicate the computational time (comparatively to the baselines, that must be recent state of the art methods), and to report the approximation error.

**Questions:**

Complementary experiments are required to assess the proposed approach and its scalability w.r.t. the dimension of the problem.
The comparison with recent baselines is mandatory.

See for instance the experimental setting in: Stochastic algorithms with descent guarantees for ICA, 2019.
Older: Consistent sparse representations of EEG ERP and ICA components based on wavelet and chirplet dictionaries, 2010.

---

### Meta-Review · Area_Chair_cgUA · 2023-12-05

**Metareview:**

The paper is interested in ICA and shows how to use a genetic algorithm to optimize the sought model.
The rebuttal addressed some comments of reviewer  ZRC8; however quite a few points (including the comparison with the references) were left unanswered. The reviewers and area chair hope that the reviews will give the authors sufficient feedback to revise their paper according to the good publication practice.

**Justification For Why Not Higher Score:**

-

**Justification For Why Not Lower Score:**

-

---

### Decision · Program_Chairs · 2024-01-16

Reject